

# Navigating post COVID-19 education: an investigative study on students' attitude and perception of their new normal learning environment

Anshoo Agarwal[1], Geetha Subramaniam[2], Osama Khattak[3], Gulam Saidunnisa Begum[4], Afaf Taha[1], Naglaa Ahmed Bayomy[5], Abdulhakim Bawadekji[6], Amin Khalid Makhdoom[7], Maali Subhi Alshammari[7] and Farooq Ahmad Chaudhary[8]

[1] Pathology Department, Faculty of Medicine, Northern Border University, Arar, Saudi Arabia
[2] INTI International University, Nilai, Malaysia
[3] Department of Restorative Dentistry, Jouf University, Sakaka, Saudi Arabia
[4] Department of Biochemistry, College of Medicine and Health Sciences, Suhar Campus, National University, Muscat, Oman
[5] Anatomy Department, Faculty of Medicine, Northern Border University, Arar, Saudi Arabia
[6] Department of Biological Sciences, College of Science, Northern Border University, Arar, Saudi Arabia
[7] Faculty of Medicine, Northern Border University, Arar, Saudi Arabia
[8] Department of Community Dentistry, Shaheed Zulfiqar Ali Bhutto Medical University, Islamabad, Pakistan

Corresponding authors
Anshoo Agarwal,
dranshoo3@gmail.com
Farooq Ahmad Chaudhary,
chaudhary4@hotmail.com

## ABSTRACT

**Background**. The incidence and aftermath of the COVID-19 pandemic brought about a drastic change in health professional education around the world. Traditional classrooms made way for online classrooms in order to ensure that learning continued in a safe and secure environment. However, how well health professional students perceived and accepted these changes have not been fully gauged yet. Therefore, this study aims to evaluate the perception of health professional students about their new educational climate.
**Methods**. A modified and validated Dundee Ready Education Environment Measure (DREEM) questionnaire was used to collect data regarding student perception of their educational environment.
**Results**. The mean DREEM scores for three time periods were in the accepted positive range of 101 to 150 indicating that most of the students perceived the changes positively. The results indicated that most students preferred blended learning over online learning or face-to-face learning alone. Areas where students were unsatisfied with their learning environment that need improvement were identified by poor item-wise scores.
**Conclusion**. Strategic remedial measures for these concerns need to be developed to improve the quality of education received by the students. However, the results of our study indicated that most of the students were able to adapt positively to the new education environment due to the change in the circumstances during COVID.

## INTRODUCTION

Health professional education, over the years, has relied on traditional teaching methods and physical classrooms to impart the necessary knowledge and skill sets to future health professional students. Nevertheless, educational institutions all around the world were forced to use e-learning platforms when COVID-19 was declared a pandemic to support students in continuing their studies (*Kaul et al., 2021*). This inevitable and sudden transformative change tested the flexibility and adaptability of teachers, students, and institutions alike. The pandemic opened several opportunities for the improvement of how medical knowledge is delivered to future health profession students. It served as a platform for the evolution of medical education (*Kaul et al., 2021*).

A primary challenge brought about by this change in the field of health professional education was the transition from traditional classrooms to online learning techniques such as live meetings and pre-recorded lectures. While educators were expected to impart the same quality and quantity of knowledge as before to students through electronic means, students were expected to understand, conform, and exhibit the expected growth in their skill sets. This posed a variety of problems (*Chaudhary et al., 2022*; *Nimavat et al., 2021*). E-learning required students to be more independent, and self-reliant and exhibit a certain amount of self-discipline to keep pace with the curriculum. Further, for health professional students the absence of clinical experience during the lockdown period was a major source of concern (*Azlan et al., 2020*) along with feelings of loneliness and depression due to isolation (*Mujtaba et al., 2023*; *Tahir et al., 2022*). Virtual classrooms limit teacher-student interaction which is an essential element in a health professional education environment. More importantly, students have been discouraged by the reduction in the practical and clinical aspects of learning during the COVID period (*Abbasi et al., 2020*; *Javed et al., 2021*). With institutions suspending regular classes, health profession students were at the risk of receiving less than an adequate amount of exposure to all the necessary spheres of knowledge and skills. This stood to weaken their performance both in exams and their ability to become skilled health professional students (*Ahmed, Allaf & Elghazaly, 2020*). To overcome this problem, simulation-based learning has been recommended to counter the absence of direct clinical experience during the pandemic (*Abdulrahman, Alamri & AlSheikh, 2022*).

Online learning had several distinct advantages too. It introduced flexibility into the students' study routine, providing them with the opportunity to learn according to their convenience. It made way for a student-oriented learning process which permitted them to study at their own pace. Students also showed a preference for recorded lectures over live classes, as they can be repeatedly listened to by students (*Kim et al., 2020*). Safety during the pandemic, cost and time conservation, and convenience have all been cited to be advantages of online education (*Hussein et al., 2020*).

The need for collecting feedback from both teachers and students is essential to understand the type of changes and improvements needed for better delivering the curriculum *via* e-learning (*Nimavat et al., 2021*). Moving forward, the future of medical education depends on how well the students, educators, and institutions adapt to the

changes. Medical education cannot be restricted to traditional learning methods as online methods be as effective as physical classrooms (*Kim et al., 2020*). The pandemic has created an opportunity to create a flexible but competent learning environment for students (*Nimavat et al., 2021*). Currently, the blended learning technique is widely considered as it encompasses both synchronous and asynchronous learning strategies. This will encourage students to be more involved in the process of their education (*Lapitan Jr et al., 2021*). The student's perception of the learning environment has been assessed globally, regionally, and to some extent locally (*Al-Naggar et al., 2014*; *Bavdekar et al., 2019*; *Palomo-López et al., 2018*). However, research papers are scarce about the comparison between the old and new normal COVID-19 learning environments in health educational institutes, especially its associations with perceptions based on gender and year of study, and academic achievement. An insight may help to analyze whether the new learning environment in which students' learning took place had been satisfying to them or not. As the educational environment strongly affects students' achievements, satisfaction, and success, it is important to gather feedback from students regarding their experience in the learning environment. Understanding the changes in perceptions of the learning environment related to student characteristics and year of training may help to give insight and help to formulate interventions that may facilitate positive experiences in education. Therefore, the importance of this study was to assess students' perceptions of the educational environment during pre-COVID, during COVID, and post-COVID and to study how it affected their academic levels and achievements. Therefore, this study aimed to compare students' perceptions of pre-COVID, during-COVID, and post-COVID pandemic learning environments. Furthermore, to compare differences in perception of the new normal post-COVID learning environment among students taking different courses at Northern Border University in Saudi Arabia.

## MATERIAL & METHODS

This descriptive cross-sectional study was conducted at Northern Border University (NBU), Arar, Saudi Arabia among the health profession students from the colleges of medicine, applied science, and nursing. The ethical approval for the study was obtained from the local committee of bioethics, Northern Border University (Ref: MED-2022-11-1371), and written consent was taken from all the recruited students after explaining to them the purpose of the study. The students were selected using a random sampling method. Students who enrolled in courses in the University after the start of the pandemic and students who passed out from their respective colleges before normal classes resumed after the pandemic were excluded from the study. Only those students were approached for this study, who were enrolled in the different courses of the University during pre COVID era and had taken courses pre-COVID, during COVID, and post-COVID era. Dundee Ready Education Environment Measure (DREEM) was employed to collect the necessary data. The perception of the students of their educational environment was analyzed before, during, and after COVID-19. DREEM analysis is a widely validated method to analyze educational environments, particularly in health professional educational institutions

(*Al-Ahmari et al., 2022*; *Miles, Swift & Leinster, 2012*). DREEM analysis was chosen for this study as it provides the possibility of identifying the strengths and weaknesses associated with each institution or country independently and taking focused remedial measures as per individual scores. A modified version of DREEM was used in this study which was verified by an expert medical education team to customize the questions concerning the COVID-19 pandemic. A pilot study of a modified version of DREEM was done among 10 students taken from the health profession colleges of NBU and it showed an adequate level of consistency (Cronbach's alpha = 0.78). DREEM inventory evaluates student perception under 5 domains constituting a total of 50 items. Items 4, 9, 13,17, 25, 35, 39, 48, and 50 are negative statements and hence were reverse scored. The 50 items are scored on a 5-point Likert scale, as follows: Strongly agree–4.0, Agree–3.0, Uncertain–2.0, Disagree–1.0, and strongly disagree - 0. For individual items, a mean score of $\geq 3.5$ is considered a true positive. Items with a mean score of $\leq 2$ indicate problem areas and concerns. A mean score of 2–3 is an item that needs can be bettered for maximum benefit to students. The responses to the questionnaire are used to generate an overall score with a maximum value of 200. Completion of the inventory will be undertaken voluntarily, and data anonymity will be maintained. Items 6, 8, 13, 37, 41, 47 and 50 were modified to reflect the aims of the present study. The five domains of student perception and the maximum possible scores for each domain are,

1. Student Perception of Learning (SPL)–12 items with a maximum score of 48
2. Student Perception of Teachers (SPT)–11 items with a maximum score of 44
3. Student's Academic Self Perception (SASP)–8 items with a maximum score of 32
4. Student's Perception of Atmosphere (SPA)–12 items with a maximum score of 48
5. Student's Social Self Perception (SSSP)–7 items with a maximum score of 28

The data obtained were analyzed using Statistical Package for Social Science (SPSS) software program (IBM SPSS v.20 Inc., Chicago Il, USA) version 23. Cronbach's alpha was used to assess the internal consistency of the instrument. The mean and standard deviation were calculated for all of the items. For each of the five domains, scores were calculated as the cumulative total of individual responses for all of the items in that domain. The Kruskal-Wallis (one-way analysis of variance) test was used to analyze the difference between the mean scores of the three different groups. The confidence level was put at 0.05 level of significance.

## RESULTS

The initial study conducted during the pre-COVID period was extended after considering the impact of COVID on health professional education. A total of 300 valid responses were received and considered for each of the three timelines. The respondents were 53.4% male and 46.6% female. The results showed an increase in the total DREEM scores during the pandemic and after when regular classes had begun as compared to the pre-COVID period. Health professional students perceived the changes brought about by the pandemic positively to a good extent.

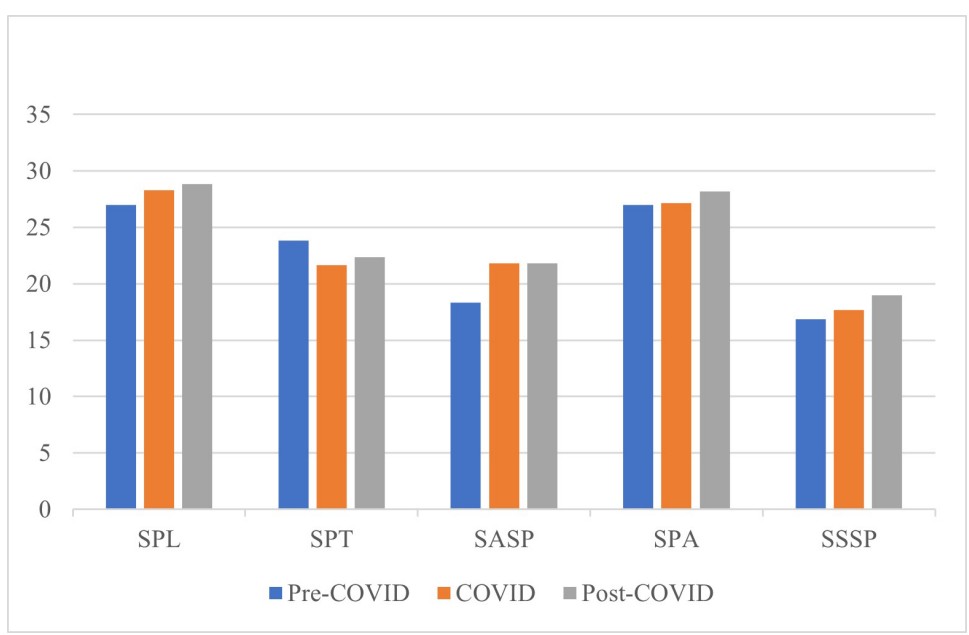

**Figure 1**  Variation of Domain scores during the three timelines.

The responses to the DREEM indicated that students perceived their educational environment positively in all three timelines and all colleges (Fig. 1).

The mean scores for individual items of the DREEM questionnaire for pre-COVID, COVID, and post-COVID periods are listed in Table 1. Of the five domains SPT alone marked a decrease in score during the COVID and post-COVID periods when compared to the pre-COVID period. The highest positive score was recorded for item 47 ('I prefer blended learning over face-to-face learning') for COVID and post-COVID durations. The lowest negative score was given to item 19 ('My social life is good') during the COVID timeline. Several problem-prone areas were identified with scores lower than 2.

DREEM questionnaire scores according to the gender of students studying in NBU colleges showed that male students responded more positively toward adaptation to the new learning environment (Fig. 2).

Table 2 lists the college-wise distribution of domain scores, and Table 3 lists the overall mean scores for each domain. Significant differences were found between the three groups for total DREEM score and the SPT, SPL, SASP, and SSSP subscale scores ($P < 0.001$).

## DISCUSSION

Due to the COVID-19 pandemic, students' learning environments had to transition to an online setting, and this had a significant impact on students' subjective happiness. DREEM has long been used to gauge the environment associated with educating medical students. It has been used as an effective tool to identify both strengths and weaknesses in other institutions (*Al-Ahmari et al., 2022*; *Miles, Swift & Leinster, 2012*). In our study too, we utilized DREEM to understand the full impact of the COVID pandemic on

**Table 1  Mean score of individual items of DREEM.**

| | DOMAINS AND ITEMS | Pre-COVID | COVID | Post-COVID | Standard deviation | P value |
|---|---|---|---|---|---|---|
| | **SPL** | | | | | |
| 1 | Encouraged to take part in classroom events | 2.2 | 2.5 | 2.5 | 1.039 | 0.005[*] |
| 2 | The educators provided thorough information regarding the course of study. | 2.5 | 2.4 | 2.6 | 1.171 | 0.001[*] |
| 3 | A good support system for stressed-out students | 2.4 | 1.9 | 2.3 | 1.014 | 0.01[*] |
| 4 | Exhausted and unable to enjoy the course | 1.8 | 2.6 | 2.6 | 1.303 | 0.006[*] |
| 5 | My approaches to learning were successful. | 2.4 | 2.2 | 2.4 | 1.046 | 0.91 |
| 6 | The course coordinators promoted a student-centered teaching approach. | 2.2 | 2.6 | 2.6 | 1.005 | 0.04[*] |
| 7 | A lesson is frequently motivating. | 2.3 | 2.7 | 2.5 | 1.041 | 0.008[*] |
| 8 | The instructors pushed active participation from the students. | 2.4 | 2.2 | 2.3 | 1.234 | 0.85 |
| 9 | The trainers are authoritarian. | 1.8 | 2.2 | 1.9 | 1.132 | 0.32 |
| 10 | I am confident that I will pass this year. | 2.5 | 2.1 | 2.3 | 1.111 | 0.12 |
| 11 | The environment remains tranquil while lecturing. | 2.4 | 2.6 | 2.6 | 1.172 | 0.88 |
| 12 | A well-planned course | 2.1 | 2.3 | 2.2 | 1.172 | 0.04[*] |
| | **SPT** | | | | | |
| 13 | There is faculty-centered teaching. | 2.2 | 1.9 | 2 | 1.28 | 0.97 |
| 14 | This course rarely makes me feel bored. | 2.1 | 2.2 | 2.1 | 1.117 | 0.07 |
| 15 | In this course of study, I have close friends. | 2 | 1.5 | 1.7 | 1.155 | 0.04[*] |
| 16 | My competency is being developed by what is being taught. | 2.3 | 2.2 | 2.1 | 1.198 | 0.08 |
| 17 | In this course, cheating is a concern. | 2 | 1.5 | 2.1 | 1.394 | <0.001[*] |
| 18 | Faculty members can effectively communicate with learners. | 2 | 1.8 | 2 | 1.076 | 0.14 |
| 19 | My social life is satisfying. | 2.2 | 1.4 | 2.1 | 1.447 | 0.45 |
| 20 | The content of the lesson seems extremely defined. | 2.3 | 2.2 | 2.1 | 1.112 | 0.02[*] |
| 21 | I believe I am being adequately prepared for my future as a professional. | 2.2 | 2.3 | 2 | 1.233 | 0.19 |
| 22 | The teaching helps to develop my confidence | 2.4 | 2 | 2.2 | 1.217 | 0.03[*] |
| 23 | During lectures, there is a comfortable environment. | 2.1 | 2.7 | 1.9 | 1.192 | 0.02[*] |
| | **SASP** | | | | | |
| 24 | The teaching time is effectively utilized. | 2.3 | 3.1 | 2.9 | 1.196 | 0.009[*] |
| 25 | The teaching above places a strong emphasis on factual learning. | 2.9 | 3 | 2.8 | 1.149 | 0.14 |
| 26 | The work from last year provided an effective foundation for this year's work. | 2.4 | 2.9 | 2.6 | 1.195 | 0.30 |
| 27 | I am able to remember any information that I need. | 2.2 | 3.1 | 2.9 | 1.14 | 0.05 |
| 28 | I don't often feel lonely. | 2.2 | 1.8 | 2.4 | 1.342 | 0.07 |
| 29 | Students receive quality feedback from the faculty members. | 1.7 | 2.9 | 2.6 | 1.222 | 0.002[*] |
| 30 | There are prospects for me to enhance my interpersonal abilities. | 2.2 | 2.1 | 2.8 | 1.21 | 0.008[*] |

**Table 1** (*continued*)

|  | DOMAINS AND ITEMS | Pre-COVID | COVID | Post-COVID | Standard deviation | *P* value |
|---|---|---|---|---|---|---|
| 31 | My work has taught me a lot about empathy. | 2.4 | 2.9 | 2.8 | 1.132 | 0.73 |
|  | **SPA** |  |  |  |  |  |
| 32 | Constructive criticism is provided by the faculty. | 2.4 | 2.8 | 2.7 | 1.212 | 0.12 |
| 33 | Socially, I feel relaxed during lectures. | 2.6 | 1.9 | 2.6 | 1.134 | 0.47 |
| 34 | During lectures, there is an informal atmosphere. / PBL | 2.1 | 2.8 | 2.3 | 1.274 | 0.11 |
| 35 | The course is disappointing to me, | 2.2 | 2.4 | 2.5 | 1.294 | 0.08 |
| 36 | I have good concentration abilities. | 2.2 | 2.3 | 2.6 | 1.109 | 0.24 |
| 37 | The tutor has access to the newest technology. | 2.4 | 1.6 | 2.3 | 1.117 | 0.93 |
| 38 | I am informed of the course's learning objectives. | 2.5 | 2.4 | 2.5 | 0.991 | 0.06 |
| 39 | The lecturers become agitated during classes. | 2.1 | 2.3 | 2.2 | 1.245 | 0.01[*] |
| 40 | The instructors for the course are well-prepared for their classes. | 2 | 1.9 | 2.1 | 1.146 | 0.07 |
| 41 | My knowledge of technology continues to develop here, which has made learning enjoyable. | 2.4 | 2.2 | 2.2 | 1.058 | 0.006[*] |
| 42 | The enjoyable aspect of the course exceeds its stress. | 1.8 | 2.2 | 1.9 | 1.266 | 0.008[*] |
| 43 | My learning is inspired by the environment. | 2.3 | 2.4 | 2.3 | 1.144 | 0.001[*] |
|  | **SSSP** |  |  |  |  |  |
| 44 | My ambition for continuous improvement is encouraged by the teacher's guidance. | 2.6 | 2.3 | 2.6 | 1.172 | 0.02[*] |
| 45 | I feel that a lot of what I must learn is related to my course. | 2.9 | 2.8 | 2.9 | 1.072 | 0.07 |
| 46 | My classroom provides an amazing environment to acquire knowledge. | 2.5 | 2.6 | 3.1 | 0.931 | <0.001[*] |
| 47 | I choose blended learning over classroom instruction. | 2.2 | 3.4 | 3.5 | 1.059 | 0.49 |
| 48 | The focus of the lesson is overly teacher-centered. | 2.1 | 2.4 | 2.3 | 1.174 | 0.70 |
| 49 | I'm convinced that I am able to ask whatever questions I want. | 2.8 | 2.1 | 2.7 | 1.206 | 0.62 |
| 50 | I support distance learning over face-to-face teaching. | 1.7 | 2.1 | 1.9 | 1.127 | 0.06 |

**Notes.**
[*]*p* < 0.05 statistically significant.

health professional students studying in different colleges of Northern Border University in Saudi Arabia. Data analysis revealed that health professional students have adapted well to their new environment, irrespective of the intensity of the changes around them. This is consistent with previous studies conducted in various universities in Sweden, Pakistan, and other countries (*Miles, Swift & Leinster, 2012*; *Syed, Faheem & Hassan, 2021*; *Vishwanathan, Patel & Patel, 2021*).

We observed an overall increase in the DREEM scores of the COVID and post-COVID timeline in comparison to the pre-pandemic period (Fig. 1). Though all three timelines reported positive scoring, this observed increase in DREEM scores during and after the pandemic can be attributed to the flexibility introduced into the otherwise tight schedule of health professional students with the introduction of online learning (*Lin, Kang & Kim, 2021*). This increase was in line with other similar studies in which health professional students aligned with the blended learning principle. Effective time management, where students may choose their own learning pace, was one of the reported advantages of online
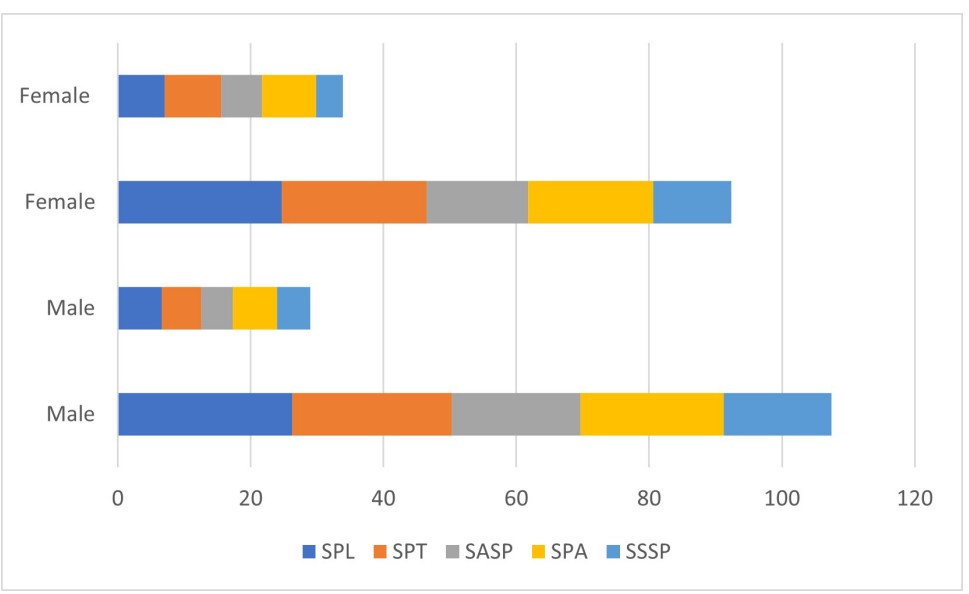

**Figure 2** DREEM questionnaire scores according to gender of students studying in NBU colleges.

learning (*Vishwanathan, Patel & Patel, 2021*). Students felt virtual classrooms to be more relaxed than physical classrooms. However, the overall positive score does not fully reveal the true picture. Several items that received ratings of less than 2 were reported as areas of concern. These areas need to be paid special attention to in order for improvement to occur.

Students' perception of learning (SPL) showed an increase in scores during and after a pandemic. In the pre-COVID period, students reported being too tired to enjoy learning (Table 1). However, the score improved during COVID when e-learning was introduced. Another area of concern in this domain was the authoritarian nature of teachers. This has been reported in other studies as well (*Vishwanathan, Patel & Patel, 2021*). *Nimavat et al. (2021)* recommended adopting both the synchronous mode (live classroom, virtual lab, *etc.*) which will allow students to immediately interact with their peers and the educator during live online sessions, and the asynchronous mode in which students can think through issues later with their classmates *via* the internet (chat rooms and discussion forums).

The only domain to report a reduction in scores during the pandemic and post-pandemic timelines was Student Perception of Teachers (SPT). The data obtained highlighted faculty-centered teaching, lack of effective communication skills of the faculty teaching, and the lack of social life as areas of concern (Table 1). During the pandemic and in its aftermath, the social lives of students were severely hindered. Students reported feeling lonely and depressed. Anxiety and frustration due to isolation and lack of communication affected a significant population of health professional students (*Aristovnik et al., 2020*; *Shahrvini et al., 2021*). In our current study too, items 19 and 28 scored negatively indicating students' mental health was impacted by the lockdown and ensuing isolation period.

**Table 2  Distribution of mean score college-wise.**

| Domain | Pre-COVID | COVID | Post-COVID |
|---|---|---|---|
| **College of Medicine (Male Campus)** | | | |
| Students' Perception of Learning | 29 | 31 | 31 |
| Student's Perception of Teachers | 22 | 20 | 22 |
| Students' academic self-perception | 18 | 21 | 22 |
| Students' perception of the atmosphere | 31 | 28 | 30 |
| Students' social self-perception | 17 | 20 | 21 |
| **TOTAL** | **117** | **120** | **126** |
| **College of Medicine (Female Campus)** | | | |
| Students' Perception of Learning | 30 | 31 | 33 |
| Student's Perception of Teachers | 24 | 23 | 24 |
| Students' academic self-perception | 16 | 19 | 18 |
| Students' perception of the atmosphere | 30 | 29 | 29 |
| Students' social self-perception | 18 | 19 | 19 |
| **TOTAL** | **118** | **121** | **123** |
| **College of Applied Science (Male Campus)** | | | |
| Students' Perception of Learning | 28 | 30 | 27 |
| Student's Perception of Teachers | 26 | 26 | 27 |
| Students' academic self-perception | 17 | 19 | 20 |
| Students' perception of the atmosphere | 29 | 24 | 30 |
| Students' social self-perception | 16 | 18 | 20 |
| **TOTAL** | **116** | **117** | **124** |
| **College of Applied Science (Female Campus)** | | | |
| Students' Perception of Learning | 23 | 24 | 23 |
| Student's Perception of Teachers | 22 | 18 | 18 |
| Students' academic self-perception | 19 | 23 | 24 |
| Students' perception of the atmosphere | 27 | 30 | 31 |
| Students' social self-perception | 16 | 15 | 18 |
| **TOTAL** | **107** | **110** | **114** |
| **College of Nursing (Female Campus)** | | | |
| Students' Perception of Learning | 27 | 28 | 30 |
| Student's Perception of Teachers | 22 | 20 | 20 |
| Students' academic self-perception | 15 | 24 | 23 |
| Students' perception of the atmosphere | 24 | 29 | 26 |
| Students' social self-perception | 17 | 18 | 19 |
| **TOTAL** | **111** | **119** | **118** |
| **College of Nursing (Male Campus)** | | | |
| Students' Perception of Learning | 25 | 26 | 29 |
| Student's Perception of Teachers | 27 | 23 | 23 |
| Students' academic self-perception | 19 | 25 | 24 |
| Students' perception of the atmosphere | 21 | 23 | 23 |
| Students' social self-perception | 17 | 16 | 17 |
| **TOTAL** | **109** | **113** | **116** |

**Table 3 Overall mean of DREEM analysis.**

| Domains | Pre-COVID | During COVID | Post-COVID | *P*-value |
|---|---|---|---|---|
| SPL | 27 | 28.3 | 28.83 | <0.001 |
| SPT | 23.83 | 21.67 | 22.33 | <0.001 |
| SASP | 18.33 | 21.83 | 21.83 | <0.001 |
| SPA | 27 | 27.17 | 28.17 | 0.057 |
| SSSP | 16.83 | 17.67 | 19 | <0.001 |
| **TOTAL** | **112.99** | **116.64** | **120.16** | **<0.001** |

The item scores in the Student's Perception of Atmosphere (SPA) domain indicated the presence of a technological gap among both students and teachers. Though this was amplified during the transition from physical to virtual classrooms. Studies indicated the reluctance of teachers to accept and make themselves familiar with the new technology platforms that were made available to aid virtual learning (*Nimavat et al., 2021*). As a result, e-learning's effectiveness was somewhat diminished. Due to a lack of technology, poor internet access, and resistance to change on the part of both students and teachers, a study in Pakistan (*Abbasi et al., 2020*) concluded that learners prefer face-to-face instruction. Just using online sessions that encompass only pre-recorded lectures and occasional face-to-face sessions may not be enough to increase student participation. *Azlan et al. (2020)* recommended improving the efficiency and scope of online education by considering collaborative educational tools and mediums. *Abbasi et al. (2020)* reported how e-teaching and learning experiences limited the number of interactions between teachers and their students. In congruence with this, the score of item 49 ('I am able to ask questions whenever I want') dropped significantly during the COVID timeline when e-learning was made exclusive for safety.

The Student's Social Self Perception (SSSP) score responses in our study provided some useful descriptive information regarding how the students perceived about distinct learning environments. The connections between the physical and psychosocial learning environments that have been found are important because maintaining a suitable learning environment becomes an ongoing challenge for educators as a poor learning environment may prevent students from learning. It is possible that a learning environment's inadequacies could also lead to a general unease that comes up on a psychosocial level, affecting the standard of the learning environment. Overall, the study reveals that students had positive perceptions of their social environment, which were reflected by relatively high levels of task orientation, cooperation, student cohesiveness, and satisfaction among them.

This study has also served the dual purpose of establishing the need for a blended learning environment for improving the educational environment of health professional students while exposing the challenges and concerns plaguing medical education. Emphasis needs to place on making the learning atmosphere stress-free by encouraging students to interact and participate more both in physical and virtual platforms. A healthy and efficient educational environment is essential for nurturing both the professional and personal
growth of future health professional students (*Aga et al., 2021*). The COVID-19 pandemic has significantly impacted medical schools around the world, with the Middle East being no exception (*Gordon et al., 2020*). Our study also showed similar results. Student's Academic Self Perception (SASP) scores reflected in our study that the outbreak made it possible for students to attend lectures or study in small groups even during online teaching. Similar findings were seen in another study when the teaching shifted online (*Rose, 2020*). To stop the spread of the infection, clinical rotations had to be put on hold until the social distance was reduced (*Chaudhary et al., 2021*; *Gaur et al., 2020*; *Kim et al., 2020*). Medical students relatively accepted the online format and were generally content with the online course (*Gaur et al., 2020*) despite the sudden transition to online learning. Although most students had favorable opinions of the online format, many of them also mentioned feeling lonely and missing their interpersonal relationships as a result of the social distance policy. Happiness is referred to as subjective well-being and is described as "a global evaluation of life satisfaction"(*Zheng, Bender & Lyon, 2021*). Considered in context with its relationship with academic performance and empathy, more recent studies have highlighted the significance of emotional well-being in health profession students (*Chaudhary et al., 2020*; *Khalil et al., 2020*). The improvement of the well-being of students as a fundamental human need and the promotion of social connectivity are both potential responsibilities of healthcare institutions (*Dworkin et al., 2021*). To further understand how health profession students' subjective happiness changes over time, it is important to look into how they perceive their educational environment (*Yoo & Kim, 2019*). A few educators have just employed the DREEM to look into the association between the learning environment and students' well-being. Positive evaluations of the medical school's learning environment were found to greatly reduce students' stress in one study (*Gordon et al., 2020*); in another, it was discovered that Student's Social Self-perceptions, one of the DREEM subscales, significantly correlated with subjective happiness (*Kim et al., 2020*). The association between this and COVID-19 has, however, barely been explored in investigations. Pre-pandemic data were not taken into consideration in the studies that already exist on health profession students' satisfaction and stress following COVID-19 (*Isaradisaikul, Thansuwonnont & Sangthongluan, 2021*; *Rose, 2020*; *Stormon et al., 2022*). Therefore, it can be difficult to figure out whether students' perspectives have changed for better or for worse following COVID-19 (*Meo et al., 2020*). As far as we understand, only a few studies have looked into how students perceived the learning environment and how happy they were both before and after the outbreak (*Villanueva, Meissner & Walters, 2021*). Our study was designed to evaluate pre-pandemic data to contrast with post-pandemic data. As a result, we could look into how COVID-19 affected health students students' views of the learning environment and their subjective happiness.

    This study has some limitations. Firstly, the survey gathers responses as students reflect on the initial lockdown session. In this context, survey response can be challenging, especially because students are urged to complete several surveys for each phase which is pre-COVID, during COVID, and post-COVID so some information may have been difficult to recall correctly. Secondly, it is questionable whether the student sample is biased in comparison to the entire cohort. Lastly, this survey was lengthy and contained 50

questions to get a complete picture of how the pandemic affected students' lives. Perhaps the survey response rate would have been higher if the survey had been designed with fewer questions and more concise question-wording.

## CONCLUSION

Students perceived the changes brought about by the pandemic positively to a good extent, however, there are emerging and evolving connections between the physical and psychosocial learning environments related to the use of new information technologies. In particular, psychosocial aspects may affect how satisfied learners are with learning in these settings. A regular training session is recommended for teachers and students alike to remove the reluctance exhibited by them toward new technologies. As teachers should be aware of students' preferences regarding the learning environment, and both its advantages and disadvantages. Education providers need to incorporate the benefits of online teaching which replaced onsite teaching due to the COVID pandemic and provide insights into a hybrid teaching, learning, and assessment approach. Further, health professional educational universities should keep in mind the mental and social health of students while designing curriculum and should establish a support system for vulnerable students. Further studies need to be done in other health educational universities to investigate variables that correlate with SPL, SPT, SASP, SPA, and SPSH and to understand the attitude of students about their environment in the wake of the pandemic.

### Funding
The authors received no funding for this work.

### Competing Interests
The authors declare there are no competing interests.

### Author Contributions
- Anshoo Agarwal conceived and designed the experiments, authored or reviewed drafts of the article, and approved the final draft.
- Geetha Subramaniam conceived and designed the experiments, authored or reviewed drafts of the article, and approved the final draft.
- Osama Khattak conceived and designed the experiments, authored or reviewed drafts of the article, and approved the final draft.
- Gulam Saidunnisa Begum conceived and designed the experiments, authored or reviewed drafts of the article, and approved the final draft.
- Afaf Taha performed the experiments, prepared figures and/or tables, and approved the final draft.
- Naglaa Ahmed Bayomy performed the experiments, authored or reviewed drafts of the article, and approved the final draft.

- Abdulhakim Bawadekji performed the experiments, prepared figures and/or tables, and approved the final draft.
- Amin Khalid Makhdoom performed the experiments, analyzed the data, prepared figures and/or tables, and approved the final draft.
- Maali Subhi Alshammari analyzed the data, prepared figures and/or tables, and approved the final draft.
- Farooq Ahmad Chaudhary analyzed the data, prepared figures and/or tables, and approved the final draft.

## Ethics

The following information was supplied relating to ethical approvals (i.e., approving body and any reference numbers):

Ethical approval for study was obtained from the local committee of bioethics, Northern Border University (Ref: MED-2022-11-1371).

## Data Availability

The raw measurements are available in the Supplementary File.

## Supplemental Information

Supplemental information for this article can be found online at http://dx.doi.org/10.7717/peerj.16184#supplemental-information.

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
