# Peer review of "Navigating post COVID-19 education: an investigative study on students’ attitude and perception of their new normal learning environment"

_PeerJ, doi:10.7717/peerj.16184_

## Round 0.1 · original submission · Major Revisions

Dear authors, in the sequence of the reviewers' comments I believe your manuscript requires some (Medium) revisions. Please, refer to the reviewers' comments for more details.

Reviewer 1 ·

Basic reporting

The manuscript is largely well written, focused and unambiguous. The language however requires some improvement in some few expressions. E.g. Line 57 :future heath professional students instead of health profession students. Line 254 : students students, etc.

Literature references are well cited and sufficient background to the objective well provided .

The structure of the article is good, figures and tables are well illustrated. However, more illustrative figures to depict the data from the table would be more appealing to the reader .

Experimental design

The research is within the aims and scope of the journal. The focus of the paper is relevant especially at a time when most universities across the globe are exploring virtual learning opportunities to provide quality education to their students.

The research question is clearly defined and clear gap well established.

The design and the measurement tools are considered appropriate. The DREEM approach has been applied in several similar studies of perception in educational institutions.

Validity of the findings

The work was found to be relevant.
However, it is quite difficult to make any judgement on the statistical soundness of the results. No description of how the data was analysed, no statistical tool? . Is there any significant difference between the DREEM scores of the studied timelines? The authors need to provide this information



Conclusions are well stated and found to be related to the research objectives. The evidence provided in the results support the conclusion drawn.

Additional comments

Lines 108 - 109 appear to suggest that the participants were selected from different universities instead of different colleges in Northern Borders University (NBU). Please kindly clarify.
Similarly, be consistent with college -wise distribution domains (Table 2) and country -wise (Line 150)

·

Basic reporting

The manuscript adresses a relevant topic.
There are space for improments that will make the manuscript more appealing to the reader.
It needs a more coherent writting and more literature review.
I kindly ask the authors to see my comments on the PDF file.

Experimental design

Regarding the aim of the study it is not clear the research question, since it was presented different aims for the study. I strongly suggest the authors to clarify this issue.
I kindly ask the authors to see my comments on the PDF file.

Validity of the findings

The conclusion should be more stronger.
I kindly ask the authors to see my commnents on the PDF file.

Additional comments

I think you could have a good manuscript, but there are a lack of coherence in your writting that needs to be urgently adressed.
Have a good work!

·

Basic reporting

This study, “Navigating Post Covid-19 Education: An investigative study on Students’ Attitude and Perception of their new normal learning environment”, is based on evaluating the perception of health professional students about their new educational climate. The descriptive and cross-sectional study used a modified and validated Dundee Ready Education Environment Measure (DREEM) questionnaire. The results are based on mean DREEM scores. The study's topic is interesting and holds good significance in the field. Following are some of my observations which would help in increasing the quality of the paper:
Minor revisions:
1. Introduction: It is well written; however, the authors should also mention the motivation behind the study along with the rationale of the study as a separate heading.
2. The literature review section needs to be included. Authors must mention previous studies conducted in the same domain and add a literature review section. A systematic literature review is required in such kinds of studies.
3. The authors should also mention the objectives of the study.

Experimental design

Materials and Method: However method section is nicely written, but it would be advisable to make some sections under it, like research methods, data collection sampling etc. The authors should also explain more about the collection of the data and pilot test of the study if conducted.

Validity of the findings

The results section needs a more detailed explanation of each research tool taken for the study. It is advisable to explain the results of mean values in line with previous studies conducted in the same domain.

Additional comments

The authors did not mention the study's implications, limitations and future research. This should be added to the manuscript.

Accepted with minor revisions.

---

## Round 0.2 · accepted · Accept

Dear authors, it is my pleasure to inform you that I and the reviewers are happy with the revisions to the manuscript and your work is now accepted for publication on PeerJ.